# Do Ophthalmic Solutions of Amphotericin B Solubilised in 2-Hydroxypropyl-γ-Cyclodextrins Possess an Extended Physicochemical Stability?

**DOI:** 10.3390/pharmaceutics12090786

**Published:** 2020-08-19

**Authors:** Philip Chennell, Mouloud Yessaad, Florence Abd El Kader, Mireille Jouannet, Mathieu Wasiak, Yassine Bouattour, Valérie Sautou

**Affiliations:** 1Université Clermont Auvergne, CHU Clermont-Ferrand, CNRS, SIGMA Clermont-Ferrand, ICCF, 63000 Clermont-Ferrand, France; ybouattour@chu-clermontferrand.fr (Y.B.); vsautou@chu-clermontferrand.fr (V.S.); 2CHU Clermont-Ferrand, Pôle Pharmacie, 63000 Clermont-Ferrand, France; myessaad@chu-clermontferrand.fr (M.Y.); florence.abdelkader@ch-lepuy.fr (F.A.E.K.); mjouannet@chu-clermontferrand.fr (M.J.); mwasiak@chu-clermontferrand.fr (M.W.)

**Keywords:** amphotericin B, γ-cyclodextrins, stability, fungal keratitis

## Abstract

Fungal keratitis is a sight-threatening disease for which amphotericin B eye drops is one of the front-line treatments. Unfortunately, there are currently no commercial forms available, and there is little data concerning the long-term stability of compounded formulations based on intravenous dosages forms. New formulations of amphotericin B ophthalmic solutions solubilised with γ-cyclodextrins have shown promising in-vitro results, but stability data is also lacking. The objective of this study was therefore to investigate the stability of a formulation of ready-to-use amphotericin B solubilised in 2-hydroxypropyl-γ-cyclodextrins (AB-HP-γ-CD), for 350 days. An amphotericin B deoxycholate (ABDC) formulation was used as a comparator. Analyses used were the following: visual inspection, turbidity, osmolality and pH measurements, amphotericin B quantification by a stability-indicating liquid chromatography method, breakdown product research, and sterility assay. AB-HP-γ-CD formulation showed signs of chemical instability (loss of amphotericin B) after 28 and 56 days at 25 °C and 5 °C. Adding an antioxidant (ascorbic acid) to the formulation did not improve stability. ABDC formulation showed signs of physical instability (increased turbidy and amphotericin B precipitation) after 28 days and 168 days at 25 °C and 5 °C. As such, AB-HP-γ-CD formulation does not provide long-term stability for ophthalmic amphotericin B solutions.

## 1. Introduction

Fungal (or mycotic) keratitis is a purulent, ulcerative infection of the cornea that can cause corneal opacification and irreversible blindness if left untreated [1,2]. It has been estimated that 1,000,000 cases occur annually in the world, but as it is often under-suspected, it is also underdiagnosed and real numbers might be a lot higher [3]. Risk factors for developing such an affection have been documented as living in a tropical or subtropical environment [4,5], underlying corneal and ocular surface diseases, ocular trauma, and wearing contact lenses [6,7,8]. Among the fungal germs isolated, the most common are Fusarium species, Aspergillus species, and Candida species [4,7,9,10,11]. For the treatment of such infections, clinicians have the choice between drugs from two main typical classes of antifungal agents that are azoles (voriconazole, fluconazole, ketoconazole, posaconazole, itraconazole) and polyenes (natamycin and amphotericin B) [12]. Of these drugs, amphotericin B is broad-spectrum agent and is active against most of fungi, especially Candida spp [13], but also possesses very low minimal inhibitory concentrations against Fusarium and Aspergillus [14], allowing it to be also one of the first line treatments of fungal keratitis caused by those germs, at dosages ranging from 0.1% to 0.5% [7,15,16,17]. Another advantage of amphotericin B is that it seems to be less likely to induce resistances as opposed to azole antifungals [18].

Amphotericin B is a heptaene possessing a heavily hydroxylated region on the ring opposite to the multiple conjugated double bonds, a mycosamine moiety and a carboxylic group (Figure 1A). These latter groups impart a polar character to the molecule (which contributes to the relative insolubility in organic solvents), whereas the opposite unsaturated terminal imparts a nonpolar character (which contributes to its poor aqueous solubility) [18]. Such properties make it hard to create an optimum formulation for ophthalmic delivery. As amphotericin B eye drops are not currently commercially available in the world, the most used formulation consists of diluting a marketed intravenous dosage form of amphotericin B deoxycholate (ABDC) in 5% glucose to reach the desired concentration. Other preparations of lipid-based formulations of amphotericin B can also be used [13,18] but they are far more expensive and not readily available in all countries. Unfortunately, ABDC is known to possess very short (less than two weeks) stability at ambient temperature, and even if longer stabilities of 60 to 120 days have been reported (often at the end point of the studies), not all physicochemical parameters were studied, making it difficult to conclude [19,20]. Overall, these shortfalls make it difficult for compounding pharmacies to adequately manage amphotericin B preparations, and short shelf life at ambient temperature means complexifying transport and storage conditions, especially for tropical and subtropical regions. To address these issues, the use of cyclodextrins to solubilize amphotericin B have been tested. Vikmon et al. first described to use of γ-cyclodextrins to solubilize up to 0.65 mg/mL of amphotericin B [21], and since then, several other authors have studied the theoretical and practical aspects of using various cyclodextrins [22,23,24], including derivatives like 2-hydroxypropyl-γ-cyclodextrin, whose chemical structure is presented Figure 1B. From what has been described, the easiest way to incorporate amphotericin B is to first dissolve the cyclodextrins in an aqueous media, then alkalinise the solution to pH 12 in order to allow the ionic form of amphotericin B to solubilize in the media and incorporate itself into the cyclodextrins. The solution is then brought back to a more tolerable pH. The complexation of amphotericin B using this preparation method has been studied by various authors and is now well documented [23,25,26]. As the first in-vitro tests performed on such formulations seem promising [27,28], it becomes important to know if cyclodextrin complexation with amphotericin B is capable to achieve ready to use formulations with long-term stability.

The objective of this study was therefore to investigate the stability of a formulation of ready-to-use amphotericin B solubilised in 2-hydroxypropyl-γ-cyclodextrins (AB-HP-γ-CD) in low-density polyethylene eyedroppers, for 350 days at 5 °C and 25 °C. A classical ABDC formulation was used as comparator.

## 2. Materials and Methods

### 2.1. Preparation of the ABDC Formulation

Amphotericin B deoxycholate powder (obtained from Fungizone^®^ powder for injectable solution vials, Bristol-Myers Squibb, Rueil-Malmaison, France) was reconstituted with sterile 5% glucose (B. Braun Medical, Boulogne Billancourt, France) to obtain a 5 mg/mL solution of AB. After complete dissolution of the powder, the solution was transferred into an empty ethylene-vinyl acetate bag (Baxter, Guyancourt, France) and diluted with a 5% glucose sterile solution to obtain a 2.5 mg/mL amphotericin B solution. All manipulations were performed under the laminar air flow of an ISO 4.8 microbiological safety cabinet.

### 2.2. Preparation of the AB-HP-γ-CD Formulation

To prepare 500 mL of 2.5 mg/mL solution of amphotericin B, initially 100 g HP-γ-CD (Wacker Chemie AG, Burghausen, Germany) were dissolved in 350 mL water for injection (WFI) (Versylène^®^ Fresenius, Bad Homburg, Germany). After total dissolution, the pH was adjusted to 12 with a 1 N sodium hydroxide solution before adding 1250 mg of amphotericin B base (pharmaceutical grade, Inresa, Bartenheim, France). The mixture was stirred to obtain a clear and orange solution then the pH was readjusted to 7.0 with 1N hydrochloric acid. The solution volume was completed to 500 mL with a NaH_2_PO_4_/Na_2_HPO_4_ phosphate buffer solution to obtain a final buffer concentration of 0.02/0.03 mol/L.

As complementary study, a formulation of 2.5 mg/mL HP-γ-CD amphotericin B eye-drops containing 0.5 mg/mL ascorbic acid (0.05%) was also prepared, as following: 100 g of HP-γ-CD was dissolved in 450 mL of WFI adjusted to pH 12 with a NaOH solution, to which 1250 mg of amphotericin B powder were solubilised. After dissolution, 3500 mg of Na2HPO4 and 2500 mg of ascorbic acid were added, and the volume adjusted to 500 mL with WFI.

### 2.3. Conditioning and Storage

The resulting solutions were sterilely conditioned (4 mL per unit) using a sterile syringe tipped with a 0.22 µm pore size filter (reference SLGP033RS, Milipore SAS, Molsheim CEDEX, France) under the laminar air flow of an ISO 4.8 microbiological safety cabinet into low density polyethylene (LDPE) eyedroppers (CAT, Lorris, France). The eyedroppers were stored at controlled refrigerated temperature (Whirlpool refrigerator) at 5 °C ± 2 °C or in a climate chamber (Binder GmbH, Tuttlingen, Germany) at 25 °C ± 2 °C and 60% residual humidity, until analysis.

### 2.4. Study Design

The stability of the different amphotericin B eye drops formulations was studied in unopened eyedroppers for up to 350 days at two different temperature 25 °C and 5 °C.

Immediately after conditioning, and then at determined times (3, 7, 14, 28, 56, 168, and 350 days after conditioning), 4 units per tested storage temperature were subjected to the following analyses: visual inspection, osmolality, pH measurements, amphotericin B quantification and breakdown products research. Turbidity measurements were performed at the same analysis times on the pooled volume from the 4 units. Additionally, for the formulation containing ascorbic acid, a chromaticity and luminescence analysis was performed.

Sterility determination assay was realized on 4 extra dedicated units after 0, 28, 168, and 350 days of storage.

### 2.5. Analyses

#### 2.5.1. Visual Inspection

The multidose eyedroppers were emptied into polycarbonate test tubes and the amphotericin B solutions were visually inspected under white light in front of a matt black panel and a non-glare white panel of an inspection station (LV28, Allen and Co., Liverpool, UK). The aspect and colour of the solutions were noted, and a screening for visible particles, haziness, or gas development was performed.

#### 2.5.2. Osmolality, pH and Turbidity Measurements

For each unit, osmolality was measured using Model 2020 osmometer (Advanced instruments Inc., Radiometer, SAS, Neuilly Plaisance, France). pH measurements were made with a SevenMultiTM pH-meter with an InLabTM Micro Pro glass electrode (Mettler-Toledo, Viroflay, France).

Turbidity was measured using a 2100Q Portable Turbidimeter (Hach Lange, Marne La Vallée, France), by the pooling of four samples per analysed experimental condition and assay time to obtain the necessary volume for the analysis. The results were expressed in Formazin Nephelometric Units (FNU).

#### 2.5.3. Amphotericin B Quantification and Breakdown Products Research

Chemicals and instrumentation

For each unit, Amphotericin B was quantified and degradation products researched using a liquid chromatography (LC). The LC system that was used was a Prominence-I LC2030C 3D with diode array detection (Shimadzu France SAS, Marne La Vallée, France) and the associated software used to record and interpret chromatograms was LabSolutions™ version 5.82 (Shimadzu France SAS, Marne La Vallée, France). The method that was used was adapted from Chang et al. [30]. The LC separation column used was a C18 a Synergi 4 µm Hydro-RP 80 Å column (Phenomenex, France). The mobile phase in isocratic mode was composed of 29.1/12.8/7.1/51 (*v*/*v*/*v*/*v*) methanol/acetonitril/tetrahydrofuran/EDTA 2.5 mM mixture. All chemicals used for the chromatography analysis were of analytical grade. The flow rate through the column for the analysis was set at 1.5 mL/min, with the column thermo-regulated to a temperature of 30 °C. The eye drops were diluted a 100-fold with deionized water, to a final concentration of 25 µg/mL. The injection volume was of 20 µL and the samples racks were kept at 20 °C. The detection wavelength for quantification was set up at 408 nm and breakdown product detection was performed using DAD detector from 190 to 800 nm.

Method validation

Linearity was initially verified by preparing one calibration curve daily for three days using five concentrations of amphotericin B (base) solubilised in dimethyl sulfoxide (DMSO) and diluted to 15, 20, 25, 30 and 35 µg/mL. Each calibration curve should have a determination coefficient R^2^ equal or higher than 0.999. Homogeneity of the curves was verified using a Cochran test. ANOVA tests were applied to determine applicability of the linear regression model. To verify method precision, six solutions of 25 µg/mL amphotericin B were prepared each day for three days, and analysed and quantified. Repeatability was estimated by calculating the relative standard deviation (RSD) of intraday analysis and intermediate precision was evaluated using RSD of inter-days analysis. Both RSDs should be of less than 5%. Specificity was assessed by comparing UV spectra obtained from the DAD detector. Method accuracy was verified by evaluating the recovery of five theoretical concentrations to experimental values found using mean curve equation, and results should be found within the range of 95–105%. The overall accuracy profile was constructed according to Hubert et al. [31,32,33]. The matrix effect was evaluated by reproducing the previous methodology with the presence of all the excipients present in the formulations and comparing the calibration curves and intercepts.

Amphotericin B impurities described in the European Pharmacopeia were either used directly from reference product (amphotericin B for peak identification CRS containing impurities A and B, catalogue code Y0001014) or were prepared (impurity B and C) following the procedure described in the Amphotericin B monography [34]. All three impurities were identified using the same method, and their retention times were collected for potential identification and quantification during stability studies.

In order to exclude potential interference of degradation products with amphotericin B quantification, 100 µg/mL amphotericin B (base and deoxycholate) solutions were subjected to the following forced degradation conditions: 0.1, 0.5 and 1N of hydrochloric acid and sodium hydroxide for 60 min at 25 °C; 10 and 30% hydrogen peroxide for 60 and 120 min; and thermal degradation at 60 °C after 1, 2 and 4 h. Susceptibility to light was performed 3 times after solution preparation after 24, 48 and 115 h of radiation exposure using UV-visible (400–800 nm wavelength, colour 640) and UVA (320–400 nm wavelength, colour 09) light. All peaks with a surface ratio higher than 0.1% of reference amphotericin B peak were taken into account for the evaluation, and those for which the surface ratio was higher than 0.2% during at least one forced degradation study were followed.

#### 2.5.4. Chromaticity Analysis

Chromaticity and luminance were measured using a UV-visible spectrophotometer (V670, Jasco France SAS, Lisses, France) using the mode Color Diagnosis of the built-in software (Spectra Manager™, Jasco France SAS, Lisses, France). The xyY CIE colorimetric system was used. Chromaticity was presented as a two dimensional diagram (x and y axes) representing the whole of the colour system independently of luminance. Luminance was defined as the visual sensation of luminosity of a surface measured by the ratio of the colour’s luminosity (in cd.cm^−2^) over the luminosity of pure white (reference colour) times 100, its value Y ranging therefore from 0 (no luminosity) to 100 (maximum luminosity).

#### 2.5.5. Sterility Assay

Sterility was assessed using the European Pharmacopeia sterility assay (2.6.1). In brief, the unidose eyedroppers were opened under the laminar airflow of an ISO 4.8 microbiological safety cabinet, and the contents filtered under vacuum using a Nalgene analytical test filter funnel onto a 47 mm diameter cellulose nitrate membrane with a pore size of 0.45 µm (ref 147-0045, Thermo Scientific, Thermo Electron SAS, Courtaboeuf CEDEX, France). The membranes were then rinsed with 500 mL of 0.9% saline solution (Versylene, Fresenius Kabi France, Louvier, France), to remove any antibacterial effect of the solution and divided into two equal parts. Each individual part was transferred to either a fluid thioglycolate medium or a soya bean casein digest medium, and incubated at 30–35 °C or 20–25 °C respectively, for 14 days. The culture medium was then examined for colonies.

### 2.6. Data Analysi–Acceptability Criteria

The stability of the different amphotericin B formulations was assessed using the following parameters: visual aspect of the solution, presence or absence of visible particles, amphotericin B concentration, presence or absence of breakdown products, pH, osmolality, and turbidity.

The study was conducted following methodological guidelines issued by the International Conference on Harmonisation for stability studies [35], and recommendations issued by the French society of Clinical Pharmacy (SFPC) and Evaluation and Research Group on Protection in Controlled Atmosphere (GERPAC) [36].

A variation of amphotericin B concentration outside the 90–110% interval of initial concentration (including the limits of a 95% confidence interval of the measures) was considered as a being a sign of significant amphotericin B concentration variation. For concentrations fluctuating between a 90–95% or 105–110% range of initial concentration, the risk of instability was assessed in regard to the presence or absence of breakdown products and the variation of the physicochemical parameters. The observed solutions must be limpid, of unchanged colour, and clear of visible signs of haziness or precipitation. Since there are no standards that define acceptable pH or osmolality variation, pH measures were considered to be acceptable if they did not vary by more than one pH unit from initial value [36]. Osmolality results were interpreted considering clinical tolerance of the preparation and turbidity measurements were considered acceptable if they did not increase by more than 10% from initial values.

## 3. Results

### 3.1. Amphotericin B Quantification and Breakdown Products Research

The retention time of Amphotericin B was of 15.49 ± 0.18 min (average ± IC95%) (Figure 2). The chromatographic method used was found linear for concentration ranging from 15 to 35 µg/mL with a mean determination coefficient R^2^ equal of 0.999. Average regression equation was y = 73,833x − 73,492 where x is the amphotericin B concentration (µg/mL) and y the surface area of corresponding chromatogram peak. Interception was not significantly different from zero.

The relative mean trueness biases were of less than 1.6%, the mean repeatability RSD coefficient was of 1.33%, and mean intermediate precision RSD coefficient was of 1.31%. The accuracy profile constructed with the data showed that the limits of 95% confidence interval coefficients were all within ±7% of the expected value (see Appendix A). The limit of detection was evaluated at 0.5 µg/mL (signal/noise ratio S/N = 21) and the limit of quantification at 5 µg/mL with S/N = 358 and a relative mean trueness of 4.1%.

Forced degradation results are presented Table 1. Amphotericin B showed high sensitivity to both acidic and alkaline conditions (degradations % ranging from 54.8% to 100%), the alkaline condition being the most aggressive, as well as to UV-visible radiations (more than 90% degradation after 24 h) and medium sensitivity to oxidation (20–25% loss after 2 h of contact with H_2_0_2_ 30%). However, amphotericin B proved quite resistant to the heat degradation, showing a loss of about 5% after 4 h at 60 °C. Breakdown products research performed with the diode array detector from wavelengths 190 to 800 nm showed that all breakdown products (18 compounds) that were detected were visible at 408 nm, none of them interfered with the amphotericin peak, and no other compounds were noticed at other wavelengths (see chromatograms provided in Appendix A). Breakdown products BP8, BP11, and BP12 were identified as being amphotericin B impurities A, B and C (see Appendix A for details). Overall, the method met all criteria for being considered as stability indicating.

### 3.2. Physicochemical Stability of ABDC and AB-HP-γ-CD Formulations

At the start of the study (day 0), the AB-HP-γ-CD formulation was a limpid amber coloured solution, whereas the ABDC formulation was a limpid yellow solution (see Appendix A
Appendix A for visual aspect images). Throughout the study, all samples maintained their initial appearance, with no appearance of any visible particulate matter, haziness, or gas development, except for the ABDC formulation stored at 25 °C, for which a haziness was noticed from day 56 onwards. This observation correlated well with the increased turbidity, raising from 11.60 FNU to 162.00 FNU at day 56, then to more than 800 FNU (maximum quantification level) after 168 days of storage. For the ABDC formulation stored at 5 °C turbidity had increased by 5.4% to 12.23 FNU after 168 days and by 124% to 26.00 FNU after 350 days. For the AB-HP-γ-CD formulation, initial turbidity was of 7.31 FNU, and decreased over time to reach 3.40 FNU (53% decrease) after 350 days when stored at 25 °C but increased over time to reach 10.40 FNU (42% increase) when stored at 5 °C.

Concerning pH and osmolality, all results are presented in Table 2. Throughout the study, osmolality did not vary by more than 15 mOsmol/kg (4.6%) and 27 mOsmol/kg (5.8%) from initial value at day 0 (320 and 465 mOsmol/kg) for respectively the ABDC and AB-HP-γ-CD formulations. pH values did not vary by more than 0.39 units except for the ABDC formulation after 336 days of storage at 25 °C, for which a decrease of 0.93 pH units was noticed, however still staying within specifications.

Amphotericin B concentrations decreased over time throughout the study, for both formulations and storage temperatures, but with wide variations between formulations and conservation temperatures (Figure 3).

After 56 days of storage, only the ABDC formulation stored at 5 °C was still within amphotericin B concentration specifications (see acceptability criteria defined in Section 2.6), and remained so up until 168 days of storage included. AB-HP-γ-CD formulation showed higher amphotericin B degradation, having lost 6.45% and 9.78% of amphotericin B after respectively 28 and 56 days when stored at 5 °C, and 18.60% and 30.0% when stored at 25 °C. For both formulations, the temperature had an important impact on degradation, as after 350 days both formulations stored at 25 °C had lost more than 60% of amphotericin B. Concerning breakdown product research, no appearance or increase of compounds already present at day 0 was detected, for neither formulation (see example chromatograms for ABDC and AB-HP-γ-CD formulations at day 168, respectively Figure 4 and Figure 5), except for the AB-HP-γ-CD formulation, for which an increase of breakdown product BP8 (impurity A) was noticed, going from 0.96% at day 0 (in % reference amphotericin B peak area) to 1.44% and 1.37% after six months and 1.69 and 0.70% after 12 months at respectively 5 °C and 25 °C.

### 3.3. Sterility Assay

None of the four analysed solutions conserved in unopened bottles at day 0, 28, 168, and 350 showed any signs of microbial growth.

### 3.4. Physicochemical Stability of AB-HP-ɣ-CD Additionned with 0.5 mg/mL Ascorbic Acid

The addition of ascorbic acid to the formulation did not modify initial visual aspect (limpid amber solution). For both conservation temperatures, turbidity, pH, and osmolality stayed within specifications (Table 3). Concentrations of amphotericin B decreased rapidly, and were out of specifications after 14 and seven days of storage when the formulations were stored respectively at 5 and 25 °C.

Interestingly, in parallel to the decrease in amphotericin B concentrations, an evolution in the colour was visually also noticed (slight darkening and reddening of the solution) with was correlated by an evolution in chromaticity and luminance measurements, which was more pronounced when the formulation was stored at 25 °C (see Figure 6).

## 4. Discussion

In this study, we show that amphotericin B solubilised in HP-γ-CD is not as stable as conventional amphotericin B deoxycholate, as amphotericin B loss reached nearly 10 and 20% after respectively 56 and 168 days of storage at 5 °C, whereas for convention ABDC formulation, the loss was of only of 6.89% and 8.70% for the same storage times.

The quantification method used in our study was adapted to our laboratory conditions from a previously published method by Chang et al. that had showed good specificity and resolution between amphotericin B and known impurities [30]. However, as it has not been validated as stability indicating, we performed accelerated degradation tests to verify the absence of interferences with potential breakdown products. The results that were obtained were coherent with those published initially by Chang et al. in terms of retention times and relative retention times, and breakdown studies allowed the detection of multiple compounds, all correctly separated from the main amphotericin B peak. They are also in accordance with the results of a very recent study by Montenegro et al. who also developed a stability indicating liquid chromatography–diode array detector method for amphotericin B quantification [37]. They confirmed that amphotericin B degrades rapidly under acidic, alkaline, oxidative and radiation exposure conditions, detecting 16 breakdown products, and made an interesting tentative shot at identifying some of the compounds using direct injection electrospray ionization mass spectrometry and electrospray ionization tandem mass spectrometry.

During our 350-day stability study, we showed that for both of the tested formulation, amphotericin B concentrations decrease over time; however, different causes could be hypothesised. For the classical deoxycholate formulation, the decrease in amphotericin B active pharmaceutical ingredient (API) happened in parallel to a massive increase in turbidity (especially at 25 °C storage), thus suggesting a physical instability rather than a chemical one, with the precipitation of amphotericin B. This is quite possibly linked to the tendency of amphotericin B to form soluble and insoluble aggregates over time, even at ambient temperature [38,39,40]. Conversely, for the AB-HP-ɣ-CD formulation, turbidity did not vary in such a way, yet API concentrations still decreased, with only a slight increase in breakdown product BP8 (impurity A), more visible at 5 °C such suggesting a certain instability of this compound at 25 °C. As the formulation was buffered at was has been described as the optimum pH range (pH 6 to 7) for amphotericin B stability [41], degradation mechanisms other than those mediated by pH are possibly implicated. Indeed, it has been suggested by several authors that amphotericin B decay can happen by autoxidation [42,43], which can also be linked to one of its antifungal activity mechanisms as it is a powerful oxidant [44,45]. Previously, Belhachemi et al. showed that adding ascorbic acid (vitamin C) and α-tocopherol (vitamin E) to amphotericin B improved its therapeutic index [46] and had hypothesised a link with lesser autoxidation, so in a complementary study we investigated if adding ascorbic acid to the AB-HP-ɣ-CD formulation might help improve its stability. Our results showed that ascorbic acid had no protective effect, and might even hasten the instability process, as after 28 days at 25 °C a 45% loss of API was noticed when in presence of vitamin C, whereas the loss was only 20% without it. The redox potential of ascorbic acid has been reported to be in the range of +0.35 V to +0.50 V [47], so it is possible that if amphotericin B red/ox potential is higher, it will not be protected by ascorbic acid. Another hypothesis could be linked to its high potential to scavenge free radicals [48], meaning that amphotericin B can also be classified as an antioxidant, with an effectiveness superior to that of retinoids but inferior to that of carotenoids [43].

Solubilization of ɣ cyclodextrin yielded a clear, very light brown solution, without any visible particulate matter. Addition of amphotericin B induced a change of color (light amber for AB-HP-ɣ-CD and light yellow for ABDC formulations) but remained limpid. Without cyclodextrins, amphotericin B is insoluble in aqueous solutions and a clearly visible precipitate is present. Other authors have confirmed that amphotericin B -ɣ cyclodextrin solution are limpid and not turbid. Rajagopan et al. observed that a minimum ratio of 1:46 (amphotericin B to ɣ cyclodextrin) was required for amphotericin B solubilisation, and that at lower ratios the solutions turned cloudy [22]. Kajtar et al. found that while in aqueous solutions amphotericin B forms colloid-like multimolecular aggregates, in the presence of γ-cyclodextrins true solutions can be prepared, which show similar spectral properties as AmB dissolved in organic solvents [25]. During preparation of topical formulations, Ruiz et al. declared that their intermediate preparation of Amphotericin B–CD inclusion complex resulted in a transparent yellow solution [27], which is also consistent with our findings. Interestingly, an evolution in colour was noted in the AB-HP-ɣ-CD formulation additionned with ascorbic acid, which was better characterized by chromaticity and luminance measurements. Indeed, chromaticity measurements clearly confirmed a slight reddening of the solution, and luminance measurements indicated that the colour was darker. Of course, it cannot be concluded at this stage which compound these modifications are related to (as ascorbic acid is known to undergo a change of colour when oxidized [49]), but these results indicate that colour measurement could be an interesting complement to be performed during stability studies of coloured solutions to track a change of colour, or of uncoloured solutions to detect any beginning coloration. Also, despite the European Pharmacopoeia for the time being only recommending the use of reference colour solutions ranging from brown to greenish-yellow [50] to assess the colouration of liquids, the United States Pharmacopoeia does have a monography describing colour measurement [51], and such a system will be implemented in the European Pharmacopoeia on the 1rst January 2021 as a harmonized text with the United States and Japanese Pharmacopoeias [52].

In 1986, Rajagopan et al. published one of the first studies describing enhanced solubility and inclusion of amphotericin B in γ-cyclodextrin complexes and proposed a mechanism for the formation of the 1:1 inclusion complex [22]. However they did not evaluate amphotericin B stability at physiological pH, only at pH 1.2 and pH 12 and for amphotericin B (base) and complexed with the cyclodextrins and for only up to 350 min. Their results did show that the amphotericin B-ɣ-cyclodextrin complex was more stable than the base amphotericin B at those extreme pH, but that amphotericin B concentrations (when complexed with ɣ-cyclodextrins) still did decrease by 10% after 350 min. No comparison with a deoxycholate formulation was made. Other authors have studied the stability of ophthalmic solutions of amphotericin B. Peyron et al. studied the stability of a deoxycholate formulation of 5 mg/mL amphotericin B diluted in dextrose, and found it precipitated after 13 to 16 days at room temperature, but declared it stable after 120 days storage at refrigerated temperature, despite not measuring the turbidity of their solution [19]. More recently, Curti et al. evaluated the physicochemical stability of five anti-infectious eyedrops, including 5 mg/mL amphotericin B, unfortunately without giving any information on the formulation (supposedly in deoxycholate form) in 5% glucose. The data they presented are also in favour of stability for 60 days at 5 °C (end point of the study), but of only 7 days at 25 °C. To the best of our knowledge, there has only been one published study investigating some stability parameters of a 0.5 and 1 mg/mL formulation of amphotericin B in γ cyclodextrins, during a short period of 30 days, at refrigerated and ambient temperatures [24]. The authors declared no API loss at either temperature when diluted in a saline solution, but noticed a 10% decrease of API when diluted in dextrose at the end of the study. At both temperatures, the antifungal activity remained unchanged. However, the study only followed the concentration of amphotericin B and particle size/aggregation effect, and the identity of the container was not mentioned, making it overall difficult to draw complete conclusions about the physicochemical stability of the formulation. The stability of liposomal amphotericin B eye drops at 5 mg/mL has also been studied [53]: the authors found that amphotericin B concentrations remained with a 94–107% range during a six-month study, and mean hydrodynamic diameter of the liposomes also remained stable, at both 5 °C and 25 °C, however one of the limits of the study was that certain parameters like pH, osmolality and turbidy were not indicated as being followed, thus making it difficult to conclude on overall formulation stability.

This study illustrates yet again that preparing adapted formulations of amphotericin B for ophthalmic use is a challenging task. The use of HP-ɣ-CD effectively allowed the preparation of concentrated amphotericin B solutions, but didn’t prevent API loss during long term storage. Recently, Jansook et al. studied the effect of additives like chitosan and phospholipids on γ-CD solubilization of amphotericin B, but have not as yet evaluated the stability of their new formulations [28]. The use of ascorbic acid as an antioxidant to prevent the loss of amphotericin B through possible autoxidation did not increase stability. Tests using different antioxidants like α-carotenoids might be a solution. Eliminating oxygen could also help reduce decay [48], but, although achievable, such a step could complexify the preparation steps, as the apparatus needed is not readily available to all compounding pharmacies. Another solution might be to try alternative excipients, such as polyethoxylated castor oil and polyvinylpyrrolidone, as they have shown to be able to solubilize other lipophilic compounds like cyclosporine A and tacrolimus [54,55]. Otherwise, and for resource-rich countries, the use of high-cost liposomal formulations could provide an acceptable, yet expensive, alternative, albeit without providing a long-term stability solution for ready-to use eye drops of amphotericin B. The search therefore continues.

## 5. Conclusions

Solutions of amphotericin B solubilised in hydroxypropyl γ cyclodextrins are not physicochemically stable for more than 28 or 56 days at 25 °C or 5 °C, respectively. Adding an antioxidant like ascorbic acid decreases the stability of the formulation. More studies are therefore needed in order to provide an affordable long-term stability solution for ready-to use eye drops of amphotericin B for the treatment of fungal keratitis caused by Fusarium species, Aspergillus species, and Candida species.

## Figures and Tables

**Figure 1 pharmaceutics-12-00786-f001:**
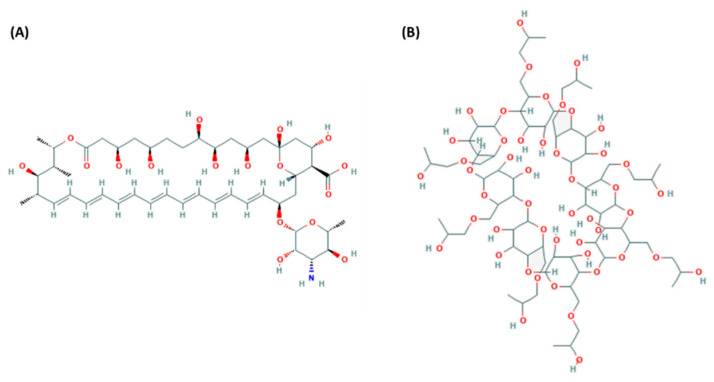
Chemical structure of amphotericin B (**A**) and 2-hydroxypropyl-γ-cyclodextrin (**B**). Publically available from [29].

**Figure 2 pharmaceutics-12-00786-f002:**
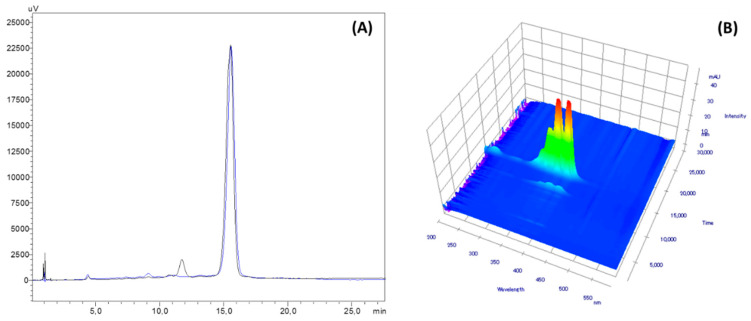
(**A**): Reference chromatogram at 408 nm of a 25 µg/mL amphotericin B base (blue curve) and deoxycholate (black curve) solution and with diode array detector screening (**B**). µV and mAU: units of intensity of signal measured by the UV-visible detector.

**Figure 3 pharmaceutics-12-00786-f003:**
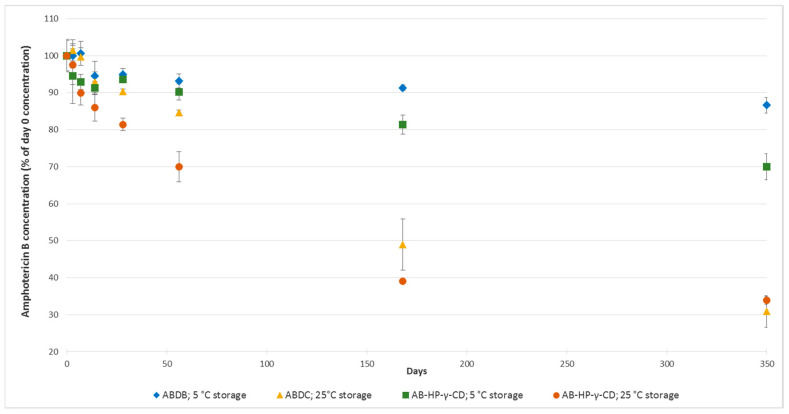
Evolution over time of amphotericin B concentrations for the amphotericin B deoxycholate (ABDC) and amphotericin B 2-hydroxypropyl-γ-cyclodextrin (AB-HP-γ-CD) formulations. *n* = 4, mean ± 95% confidence interval.

**Figure 4 pharmaceutics-12-00786-f004:**
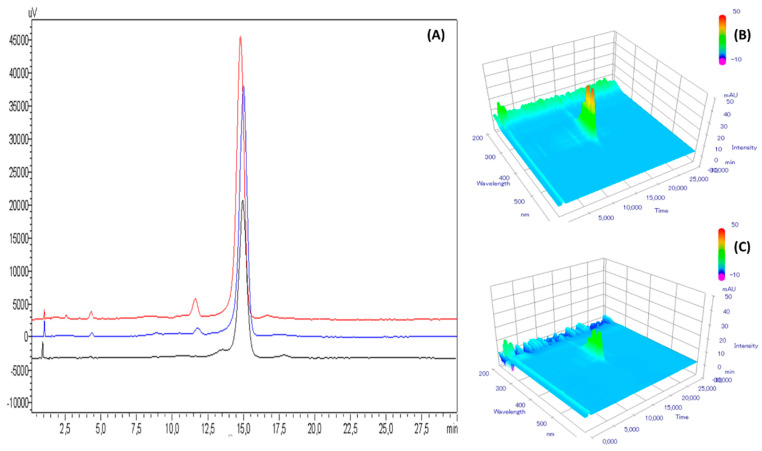
(**A**) Chromatograms at 408 nm of amphotericin B deoxycholate (diluted 1/100th to the theoretical concentration of 25 µg/mL) at day 0 (red curve), after 168 days at 5 °C storage (blue curve) and 25 °C (black curve); with diode array detector screening after 168 days at 5 °C (**B**) and at 25 °C (**C**). µV and mAU: units of intensity of signal measured by the UV-visible detector.

**Figure 5 pharmaceutics-12-00786-f005:**
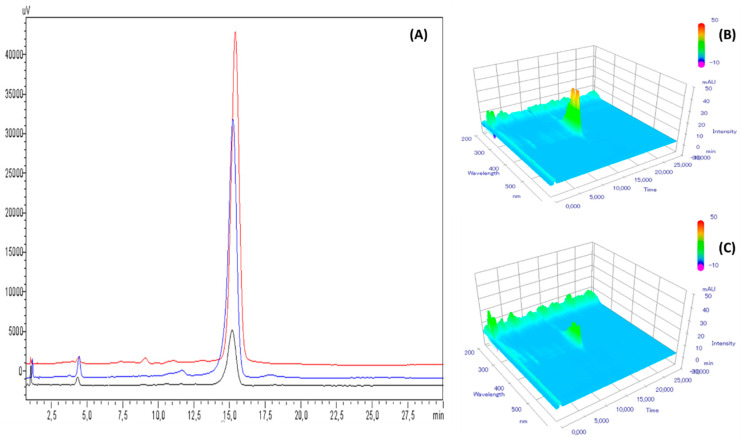
(**A**) Chromatograms at 408 nm amphotericin B solubilised in 2-hydroxypropyl-γ-cyclodextrins (diluted 1/100th to the theoretical concentration of 25 µg/mL) at day 0 (red curve), after 168 days at 5 °C storage (blue curve) and 25 °C (black curve); with diode array detector screening after 168 days at 5 °C (**B**) and at 25 °C (**C**). µV and mAU: units of intensity of signal measured by the UV-visible detector.

**Figure 6 pharmaceutics-12-00786-f006:**
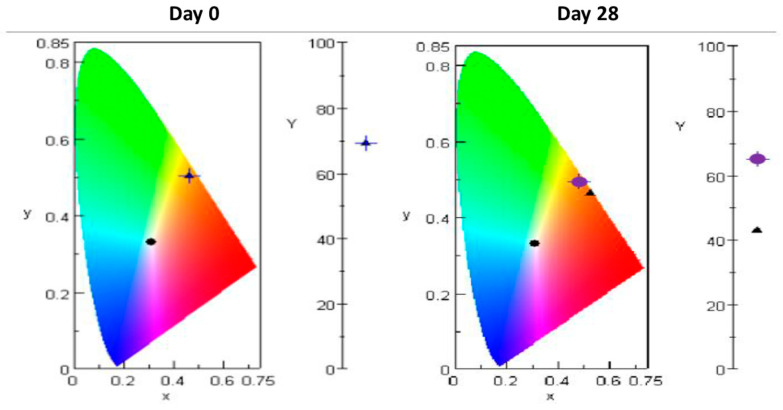
Chromaticity (xy diagram) and luminance (Y) results of the HP-γ-CD amphotericin B formulation containing 0.5 mg/mL ascorbic acid at day 0 and day 28. Blue cross: day 0 results. Purple cross target: storage at 5 °C. Black triangle: storage at 25 °C.

**Table 1 pharmaceutics-12-00786-t001:** Amphotericin B forced degradation results for different conditions, in % of reference amphotericin B peak area. BP: Breakdown product. RRT: relative retention time compared to amphotericin B retention time.

Amphotericin B (Base)																		
Compound	RRT Mean	Reference	60 °C			HCl 1 h Contact			NaOH 1 h Contact			H_2_O_2_ 10%		H_2_O_2_ 30%			With UV	
			1 h	2 h	4 h	0.1 N	0.5 N	1 N	0.1 N	0.5 N	1 N	1 h	2 h	1 h	2 h	24 h	48 h	115 h
BP1	0.07																	0.1%
BP6	0.17						0.2%				0.1%							
BP7	0.23			0.1%	0.1%													
BP8	0.29	0.5%	0.3%	0.2%	0.2%	0.5%	0.9%			0.1%		0.4%	0.5%	0.3%	0.2%	0.1%		
BP9	0.37	0.1%		0.2%	0.1%								0.1%	0.6%	0.8%			
BP10	0.49	0.1%		0.2%	0.2%		0.1%									0.3%	0.1%	
BP11	0.54		0.2%	0.3%	0.3%					0.1%		0.1%				0.0%		
BP12	0.59	0.9%	0.6%	0.6%	0.6%	0.7%					0.2%	0.1%				0.1%		
BP13	0.70	0.6%				0.8%	0.3%	0.2%	0.5%	0.1%		1.1%	2.3%	11.3%	17.5%	0.1%		
BP14	0.77		0.1%	0.4%	0.8%		2.5%	0.2%		0.1%		2.5%	2.1%	1.1%	0.6%			
BP15	0.85			0.3%	0.3%	0.8%	0.6%	0.2%										
Amphotericin B		100.0%	97.5%	96.4%	94.6%	30.1%	14.8%	1.3%	5.3%	0.9%	0.3%	91.9%	91.6%	81.3%	74.9%	5.2%	0.8%	0.0%
BP16	1.13											1.1%	1.2%	0.3%	0.8%			
BP17	2.21												0.9%	0.7%	0.9%			
BP18	2.59												1.1%	0.8%	1.4%			
**Amphotericin B (Deoxycholate)**																		
**Compound**	**RRT Mean**	**Reference**	**60 °C**			**HCl 1 h Contact**			**NaOH 1 h Contact**			**H_2_O_2_ 10%**	**H_2_O_2_ 30%**			**With UV**	
			**1h**	**2h**	**4h**	**0.1 N**	**0.5 N**	**1 N**	**0.1 N**	**0.5 N**	**1 N**	**1 h**	**2 h**	**1 h**	**2 h**	**24 h**	**48 h**	**115 h**
BP1	0.07															0.5%	0.5%	0.2%
BP6	0.17		0.1%	0.1%	0.1%				5.8%	46.6%	43.4%					0.1%		
BP7	0.23		0.1%	0.1%	0.1%					0.3%	0.3%							
BP8	0.29	0.5%	0.2%	0.2%	0.2%	0.2%	1.5%	1.9%	0.1%					0.2%	0.2%	0.4%	0.4%	0.1%
BP9	0.37									0.1%	0.1%	0.1%		0.4%	0.7%			
BP10	0.49		0.2%	0.2%	0.3%													
BP11	0.54		0.3%	0.4%	0.5%			0.1%										
BP12	0.59	0.2%	0.5%	0.6%	0.6%		0.4%	0.2%								0.3%		
BP13	0.70	0.7%	0.7%	0.7%	0.6%	0.3%	0.6%	0.2%				1.3%	2.1%	9.9%	15.8%	0.6%		
BP14	0.77	6.0%	4.6%	4.1%	3.6%	1.7%	3.9%	7.6%	8.4%	0.2%		1.0%	1.0%	0.6%	0.3%	5.3%	2.6%	0.7%
BP15	0.85		0.2%	0.2%	0.3%		0.3%	0.3%				0.2%	0.3%	0.3%	0.3%			
Amphotericin B		100.0%	97.7%	97.3%	96.0%	45.2%	36.1%	30.9%	33.8%	0.9%	0.0%	93.6%	93.1%	85.8%	79.3%	6.5%	1.8%	0.0%
BP16	1.13						0.9%	0.2%				1.0%	1.0%	0.6%	0.8%			
BP17	2.21												0.7%	0.6%	0.5%			
BP18	2.59												1.4%	0.9%	0.9%			

**Table 2 pharmaceutics-12-00786-t002:** Evolution over time of pH and osmolality for the amphotericin B deoxycholate and amphotericin B 2-hydroxypropyl-γ-cyclodextrin formulations. *n* = 4, mean ± standard deviation. Osmolality in mOsmol/kg.

Storage Time (Days)	Amphotericin B Deoxycholate				Amphotericin B 2-Hydroxypropyl-γ-Cyclodextrin			
	5 °C Storage		25 °C Storage		5°C Storage		25°C Storage	
	pH	Osmolality	pH	Osmolality	pH	Osmolality	pH	Osmolality
0	7.64 ± 0.02	320 ± 1	7.64 ± 0.02	320 ± 1	7.14 ± 0.02	465 ± 11	7.14 ± 0.02	465 ± 11
3	7.65 ± 0.00	323 ± 1	7.60 ± 0.05	320 ± 1	7.12 ± 0.01	458 ± 4	7.14 ± 0.02	463 ± 9
7	7.63 ± 0.01	320 ± 2	7.59 ± 0.00	321 ± 3	7.00 ± 0.00	457 ± 3	6.99 ± 0.00	458 ± 9
14	7.61 ± 0.01	321 ± 1	7.51 ± 0.01	320 ± 0	7.08 ± 0.01	471 ± 9	7.06 ± 0.02	457 ± 7
28	7.57 ± 0.00	322 ± 3	7.42 ± 0.02	320 ± 1	7.06 ± 0.00	456 ± 12	7.04 ± 0.00	463 ± 6
56	7.55 ± 0.01	325 ± 3	7.35 ± 0.00	326 ± 1	7.06 ± 0.01	466 ± 9	7.02 ± 0.00	465 ± 2
168	7.44 ± 0.01	323 ± 3	7.02 ± 0.01	327 ± 1	7.02 ± 0.00	468 ± 27	6.93 ± 0.01	468 ± 13
350	7.25 ± 0.01	335 ± 1	6.71 ± 0.01	330 ± 4	6.96 ± 0.00	492 ± 31	6.81 ± 0.00	478 ± 60

**Table 3 pharmaceutics-12-00786-t003:** Evolution of studied parameters for the amphotericin B 2-hydroxypropyl-γ-cyclodextrin formulation added with 0.5 mg/mL of ascorbic acid. *n* = 4; mean ± standard deviation, except for *: *n* = 1. FNU: Formazin Nephelometric Units.

		Turbidity (FNU) *	pH	Osmolality (mOsmol/kg)	Concentration (mg/mL for Day 0 Then % of Day 0 Concentrations)
Before storage	Day 0	3.19	7.08 ± 0.00	423 ± 5	2.47 ± 0.05
Storage at 5 °C	Day 7	3.20	7.09 ± 0.03	452 ± 4	96.54 ± 3.10
	Day 14	3.14	6.95 ± 0.02	449 ± 22	90.09 ± 2.31
	Day 28	3.39	6.79 ± 0.01	450 ± 11	83.93 ± 1.21
Storage at 25 °C	Day 7	3.19	6.84 ± 0.00	453 ± 12	85.08 ± 1.58
	Day 14	3.23	6.64 ± 0.02	479 ± 4	72.41 ± 3.81
	Day 28	3.32	6.38 ± 0.06	468 ± 7	55.32 ± 8.96

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
