# Peer review of "Do Ophthalmic Solutions of Amphotericin B Solubilised in 2-Hydroxypropyl-γ-Cyclodextrins Possess an Extended Physicochemical Stability?"

_pharmaceutics, 2020, doi:10.3390/pharmaceutics12090786_

Round 1

Reviewer 1 Report

The chemical structures of amphotericin B and 2-hydroxypropyl-γ-cyclodextrin should be given.

The manuscript contains some misprints and inaccuracies. The following corrections should be inserted in the manuscript.

Page 1, line 26. “…does not provided long-term stability”

Possible correction: does not provide long-term stability

Page 4, line 138. Chang et al [25].

Correction: Chang et al. [25].

Page 4, line 146. 190 to 800 nm

Correction: 190 to 800 nm.

Page 4, lines 159-160. Hubert et al [26–28].

Correction: Hubert et al. [26–28].

Page 4, line 161. and intercepts

Correction: and intercepts.

Page 4, line 166. and their retention time were collected

Possible correction: and their retention times were collected

Page 4, lines 175-177. Please check the sentence: “All peaks with a surface radio higher than 0.1% of reference amphotericin B peak were taken into account for the evaluation, and those for which the surface ration was higher than 0.2% during at least one forced degradation study were followed.” (please check: a surface radio; the surface ration).

Page 4, line 184. luminosity (in cd.cm-2)

Correction: luminosity (in cd·cm-2)

Page 5, lines 218-219. “The retention time of Amphotericin B was of 15.49 ± 0.18 min (average ± IC95%) (Error! Reference source not found.).”

Page 6, Figure 1A. What does uV on the ordinate axis mean?

Figure 1B, diode array detector screening. What does mAV on the ordinate axis mean?

Page 6, line 234. “Forced degradation results are presented Error! Reference source not found..”

Page 7, Table 1. H2O2 should be replaced by H2O2.

Page 7, line 252. The phrase “At T0,” should be explained.

Page 7, lines 256-257. from 11.60 FNU to 162.00 at day 56, then to more than 800

Correction: from 11.60 FNU to 162.00 FNU at day 56, then to more than 800 FNU

Page 7, line 262. “Concerning pH and osmolality, all results are presented in Error! Reference source not found..”

Page 8, line 276. “…temperatures (Error! Reference source not found.).”

Page 8, Figure 2. What does T0 on the ordinate axis mean?

Page 8, lines 290-291. “…γ-CD formulations at day 168, respectively Error! Reference source not found. and Error! Reference source not found.),”

Page 8, line 294. 5°c and 25°C .

Correction: 5°C and 25°C.

Page 9, Figures 3A and 4A. What does uV on the ordinate axis mean?

Figures 3B, 3C, 4B and 4C. What does mAV on the ordinate axis mean?

Page 9, line 298. “with diode array detector screening (B) after 168 days at 5°C and (C) at 25°C.”

Possible correction: with diode array detector screening after 168 days (B) at 5°C and (C) at 25°C.

Page 9, lines 302-303. “with diode array detector screening (B) after 168 days at 5°C and (C) at 25°C.”

Possible correction: with diode array detector screening after 168 days (B) at 5°C and (C) at 25°C.

Page 10, lines 309-310. “For both conservation temperatures, turbidity, pH and osmolality stayed within specifications (Error! Reference source not found.).”

Page 11, lines 332, 336. Chang et al

Correction: Chang et al.

Page 11, line 339. Montenegro et al

Correction: Montenegro et al.

Page 11, line 347. What does API mean?

Page 12, line 358. Belhachemi et al

Correction: Belhachemi et al.

Page 12, line 382. Peyron et al

Correction: Peyron et al.

Page 12, line 386. Curti et al

Correction: Curti et al.

Page 12, line 406. Jansook et al

Correction: Jansook et al.

Page 13, lines 435-436. Ref. 2. Mycotic Infections of the Eye.

Correction: Mycotic infections of the eye.

Analogous corrections should be made in References 3, 6, 7, 8, 10, 14, 18, 32, 33, 34, 42, 51.

Page 13, line 438, Ref. 3. J. Fungi 2017, 3,

Correction: J. Fungi 2017, 3, 57.

Page 13, line 444, Ref. 5. 2020, 19,

Correction: 2020, 19, 11

Page 13, line 450, Ref. 7. 2020, 10,

Correction: 2020, 10, 133

Page 14, lines 482-483, Ref. 21. SOLUBILIZATION OF AMPHOTERICIN B WITH γ-CYCLODEXTRIN.

Correction: Solubilization of amphotericin B with γ-cyclodextrin.

Page 15, line 535, Ref. MedChemComm

Correction: Med. Chem. Commun.

Reviewer 2 Report

This manuscript thoroughly describes a test of stability of several ophthalmic formulations of amphotericin B. The described work is thorough and well-validated, and describes valuable and practical information on storage of AmB formulations. Fig. 2 is a practical and important presentation of key data.

There are several minor corrections needed.

Figures are not properly numbered or referenced in the text. This may be related to the recurrence of an error message in the text “(Error! Reference source not found.)” See also mis-numbering of the final figure.

Line 224: should be “…bias was…” or …”biases were…”

Table 1: spelling of “amphotericin” should be uniform with or without the final “e”

Line 289 should be “either”

Line 359 should be “its” rather than “it’s.”

For readers not familiar with pharmaceutical nomenclature, should “API” be defined?

Supplemental Fig. 2:

“radiation” is mis-spelled

The label “4H” in D needs to be explained in the figure legend

The word “exposition” should be replaced with “exposure” throughout

Identification of amphotericin B impurities: “European” is misspelled

Supplemental data table: Row 12 should say “Amphotericin B deoxycholate, 25°C storage.”

Reviewer 3 Report

This paper report the inefficacy of cyclodextrin (CD) in the stabilization of amphotericin B for its long-term use. In many studies, the formation of inclusion-complexation between CD and guest molecules showed an increase in the stability of the encapsulated molecules during their storage. However, in this paper, the authors found that the presence of CD did not improve the long-term stability for ophthalmic amphotericin B solutions. Even such solutions were less stable than amphotericin B deoxycholate solutions. Maybe there is no inclusion-complexation (IC) between them, so they could not observe an increase in their stability. According to the study of Rajagopan et al., the complex of amphotericin-B with γ-CD enhanced amphotericin-B stability in solution over free amphotericin-B (doi.org/10.1016/0378-5173(86)90113-4). The authors first had to confirm the formation of IC between HP-g-CD and amphotericin-B. Otherwise, the presence of CD in the amphotericin-B solution will not affect its stability. Overall the paper relies on unsuccessful attempt to increase the stability of amphotericin-B solutions. Whereas the authors should able to increase its stability using another technique to able to publish in this journal. For example, Morand et al. reported the liposomal amphotericin B eye drops to treat fungal keratitis, and they found that the liposomes retained their amphotericin B content throughout the six months of the study (10.1016/j.ijpharm.2007.04.028).

Furthermore, the manuscript was carelessly prepared for submission. There are many citation issues "Error! Reference source not found. The figures are poor quality and no easy to read through. The font sizes in some figures are too small, making them impossible to read the axis of some figures...

Because of the above reasons, I cannot recommend this paper for publication in Pharmaceutics.  

Other points;

  • Lines 253-254 "Throughout the study, all samples maintained their initial appearance, with no appearance of any visible particulate matter, haziness.." The solution of amphotericin-b and HP-g-CD is transparent, with no haziness? Generally, CD IC solutions look turbid. The formation of IC should be confirmed before assessing its effect on stability. The authors claimed an increased solubility of amphotericin-b with CD.. This should be confirmed by IC formation tests.
  • The authors used 5% glucose solution for amphotericin-B. There might be competition between glucose and amphotericin-B. There are some studies showing the interaction between CDs and glucose (Can. J. Chern. 73, 12 (1995)). In the paper, they used a substantial amount of glucose compared to amphotericin-B. In the case of any interactions between the CD and glucose (does not need to be host-guest complexation), the CD cavity can be hindered. I suggest some additional experiments to confirm the host-guest interaction between CD and amphotericin-B using 1H- and 2D-ROESY NMR spectroscopy techniques.
  • The conclusion part should be extended.
  • Line 77 "Amphotericin B deoxycholate powder (obtained from Fungizone®powder for injectable solution vials, Bristol-Myers Squibb, Rueil-Malmaison, France)". Line 85, amphotericin B was purchased from where? What is its purity?
  • The font size in some figures is too small, making it impossible to read (e.g., Fig. 1b, Fig. 3b, Fig. 4B)
  • Page numbers are missing in the Supporting information file
  • Line 310, missing ref.
  • Ref 21 should be corrected.
  • Ref 30 is not accessible. It should be updated.
  • Ref 32, the authors' names were written twice.. it should be corrected.
  • Throughout the manuscript, there are problems with citations "Error! Reference source not found" They should be corrected.

Round 2

Reviewer 3 Report

I cannot still see any experimental proof of inclusion-complexation between CD and amphotericin B. Whereas, the authors cited some papers that claim the IC formation between them. Since the paper claims the inefficacy of the CD in the stabilization of amphotericin B for its long-term use, the formation of inclusion-complexation has to be shown experimentally..